# A comparation of dexmedetomidine and midazolam for sedation in patients with mechanical ventilation in ICU: A systematic review and meta-analysis

Jiaxuan Wen[1], Xueying Ding[1☯], Chen Liu[1], Wenyu Jiang[2], Yingrui Xu[1], Xiuhong Wei[1], Xin Liu[3]*

1 School of Nursing, Weifang Medical University, Weifang, P. R. China, 2 School of Public Health, Weifang Medical University, Weifang, P. R. China, 3 Department of Neonatology, Weifang People's Hospital, P. R. China

☯ These authors contributed equally to this work.
* xinliuwf@163.com

**Data Availability Statement:** All relevant data are within the paper and its Supporting Information files.

## Abstract

### Background

The use of dexmedetomidine rather than midazolam may improve ICU outcomes. We summarized the available recent evidence to further verify this conclusion.

### Methods

An electronic search of PubMed, Medline, Embase, Cochrane Library, and Web of Science was conducted. Risk ratios (RR) were used for binary categorical variables, and for continuous variables, weighted mean differences (WMD) were calculated, the effect sizes are expressed as 95% confidence intervals (CI), and trial sequential analysis was performed.

### Results

16 randomized controlled trials were enrolled 2035 patients in the study. Dexmedetomidine as opposed to midazolam achieved a shorter length of stay in ICU (MD = -2.25, 95%CI = -2.94, -1.57, p<0.0001), lower risk of delirium (RR = 0.63, 95%CI = 0.50, 0.81, p = 0.0002), and shorter duration of mechanical ventilation (MD = -0.83, 95%CI = -1.24, -0.43, p<0.0001). The association between dexmedetomidine and bradycardia was also found to be significant (RR 2.21, 95%CI 1.31, 3.73, p = 0.003). We found no difference in hypotension (RR = 1.44, 95%CI = 0.87, 2.38, P = 0.16), mortality (RR = 1.02, 95%CI = 0.83, 1.25, P = 0.87), neither in terms of adverse effects requiring intervention, hospital length of stay, or sedation effects.

### Conclusions

Combined with recent evidence, compared with midazolam, dexmedetomidine decreased the risk of delirium, mechanical ventilation, length of stay in the ICU, as well as reduced

**Funding:** The author(s) received no specific funding for this work.

**Competing interests:** The authors have declared that no competing interests exist.

patient costs. But dexmedetomidine could not reduce mortality and increased the risk of bradycardia.

## Introduction

Sedation is often required for critically ill patients in the intensive care unit (ICU) to reduce patient-ventilator asynchrony and minimize delirium and agitation for comfort [1, 2]. Conventional sedatives include benzodiazepines, dexmedetomidine, propofol, and opioids [3], of which benzodiazepines are gamma aminobutyrate receptor agonists of the central nervous system and have anti-anxiety, amnesia, sedation, hypnosis, and anticonvulsant effects [4]. As the most common drug in this class, midazolam has the characteristic of rapid onset and short duration [5]. As a selective Alpha-2 adrenergic receptor agonist, by inhibiting the release of norepinephrine from the nucleus coeruleus and competitively binding to the Alpha-2 receptor, dexmedetomidine plays a role in relieving calming, antianxiety and mild analgesic sedation [6].

Although the recent international guidelines offered a conditional recommendation favoring the use of dexmedetomidine over benzodiazepines in mechanically ventilated adults [3], in real clinical practice, midazolam is still frequently used [7–10], especially when the COVID-19 pandemic caused a shortage of medicines [11]. Besides, this recommendation was based on studies published between 2007 and 2015 [12, 13], and the quality of some of these articles was questionable. Because of the low level of evidence, international guidelines issued only a conditional recommendation [3, 14]. Some previous analysis suggested that dexmedetomidine provided an advantage over midazolam in terms of shorter sedation duration and fewer adverse events over midazolam, the low quality of inclusion articles [15] or few outcome measures [16] prompted us to update our analysis. In addition, although previous studies confirmed that dexmedetomidine was superior to benzodiazepines for mechanical ventilation in the ICU [17, 18], these previous researches included studies comparing dexmedetomidine to lorazepam. As a medium-effect benzodiazepine central nervous system inhibitor, the clinical pharmacology of lorazepam is significantly different from that of midazolam. Even short-term injection of lorazepam for sedation may lead to significant delays in extubation and discharge of ICU patients [19]. Thus, this comparison may overstate the adverse effects of midazolam.

Based on the latest evidence, we attempted to expand the previous analysis to comprehensively analyze the efficacy and adverse effects of the two sedatives. Moreover, we attempted to determine whether the results reached the size of the information required to draw conclusions in conjunction with the trial sequence analysis.

## Materials and methods

We followed the PRISMA statement and the guidelines in the Cochrane Handbook for reporting systematic reviews and meta-analyses in order to ensure the reliability and authenticity of the results. [20, 21]. Our meta-analysis had been registered with the PROSPERO database with registration number CRD42022379612. The PRISMA checklist is shown in S1 Table.

### Electronic search strategy

A search of the databases Medline, Embase, PubMed, Cochrane Library, Web of Science, EBSCO and Scopus was performed by two researchers. The search identified relevant articles without language restriction from inception to October 15th, 2023. During the electronic

database search, Among the MeSH terms and keywords were "dexmedetomidine", "midazolam", "mechanical ventilation" and "intensive care". To obtain additional trials of high relevance and to ensure the accuracy of subsequent analysis, we filtered the additional research in the reference list of relevant articles.

## Eligibility criteria

Only randomized controlled trials published as full articles were included without language restrictions. These randomized controlled trials adhered to the following criteria: (a) patients over the age of 18, (b) admitted to the ICU requiring mechanical ventilation, (c) clinically needed sedation, (e) and received dexmedetomidine in the intervention group and midazolam with or without other sedatives in the control group. (f) trials with outcome indicators included: delirium, intensive care unit length of stay, duration of mechanical ventilation, hospital length of stay, all-cause mortality, proportion of time at target sedation, additional sedatives use, and adverse events including bradycardia, hypotension, hypertension, tachycardia.

## Data extraction and analysis

Data were extracted and cross-checked by two researchers after reviewing the literature independently. If there was a disagreement, it should be resolved through discussion or consultation with a third reviewer. A designed data form was used to extract and manage data from all qualified literature. We collected data on title, author, year of publication, patient age, gender, number of cases, acute physiology and chronic health evaluation (APACHE) II, target sedation, drug dose and outcome indicators. Trials that report a continuous outcome as a median and dispersion were converted into a mean and standard deviation [22, 23]. To obtain missing data from a study, the original author was contacted via phone or email.

## Literature quality assessment

As per the recommendations in the Cochrane Handbook for the Systematic Review of Interventions 5.1.0 [24], two reviewers independently and in duplicate assessed trials for bias. We resolved disagreements through discussion and consensus. Among the contents assessed are the randomization method, the allocation concealment design, the blinding methods used, integrity of the results report, selective reporting of outcome and the existence of other sources of bias, etc. Following are the results: An answer of "Yes" indicated correct methodology or complete data, indicating a low risk of bias; "unclear" indicated a medium risk of bias; and "No" indicated a high risk of bias. The article was considered low risk of bias if the bias in each domain was low, otherwise it was regarded as high risk. A risk of bias assessment plot was exported using RevMan 5.4 software.

## Statistical analysis

Revman5.4 was used to conduct data analyses. During the meta-analysis, we used the DerSimonian and Laird random-effect models [25]. For dichotomous outcomes, we calculated pooled relative risks (RR), for continuous outcomes, we calculated mean differences (MD), and we calculated corresponding confidence interval (CI) of 95%. In order to assess the statistical heterogeneity, the I2 index was calculated, low heterogeneity was defined as I2<50% [26]. Publication bias was checked through visual inspection of funnel plots and by Egger's or Egger's tests using STATA software V.16.0. [27, 28].

In order to examine whether baseline factors influenced treatment effects and explore heterogeneity between trials, we performed a subgroup analysis. Subgroup analyses were

conducted for the outcome by age of participants, region, as well as sedation duration. We conducted meta-regression analyses in Stata 16.0 to examine the relationship between delirium and the mean age of patients. To evaluate whether the results were robust, we performed a sensitivity analysis by excluding studies with small sample sizes (n<100) or studies with a high risk of bias.

To determine if the required sample size for statistical significance was met, we performed trial sequential analysis (TSA) using TSA software v. 0.9.5.10 Beta [29]. A conclusion could be considered firmly accepted or refuted when the required information size (RIS) was larger than accrued information size (AIS) or the cumulative z-curve crossed the sequential monitoring boundary. Otherwise, further trials were needed to verify the results. The type-I error (Alpha-2) and power were set as 0.05 and 0.80, respectively. The relative risk reduction in analysis of All dichotomous outcomes is 30%, for continuous data, the "empirical" item or minimal difference was set for MD and variance, and the "model-based variance" item was applied.

GRADE (Grading of Recommendations Assessment, Development, and Evaluation) was used by two reviewers in evaluating the quality of evidence [30]. GRADEpro software was used to estimate the quality of our results on the basis of risk of bias, inconsistency, indirectness, imprecision, and publication bias. The quality of the results was categorized into four grades: high, moderate, low, or very low [31].

## Results

### Search results

A preliminary search identified 788 citations (Pubmed:42; Embase:124; Cochrane Library:86; Web of Science:109; EBSCO:307; Scopus:120), and 549 records remained after the removal of duplicate records. We screened the titles and abstracts. A further 524 records were removed after that. The full-text of 25 articles was carefully read for eligibility. The full-text of 9 articles was excluded for a variety of reasons. Finally, 16 trials were included in our analysis. The overview of the search strategy is shown in the S2 Table in S1 File. Fig 1 represents the PRISMA flow chart of the search process.

### Study characteristics

Overall, 16 eligible trials enrolled a total of 2035 patients. Table 1 describes the baseline characteristics of included literature. These trials were published between 2009 and 2022. Study participants ranged from 23 to 500. In all cases, patients required mechanical ventilation in the intensive care unit. The overall mean age of patients was 60.52±15.99 years. An estimated 40.3% of participants were female. And the mean APACHE II score reported in 14 trials was 19.88±7.28. 9 of the studies were conducted in Asian countries, three in the United States, three in European countries and one in Australia. 11 trials showed sedation lasting longer than 24 hours. A maximum rate of 0.7μg/kg/h was used for infusion of dexmedetomidine in most trials, and the infusion rate of midazolam varied greatly among different trials. All the trials maintained a light level of sedation although the evaluation methods were different.

### Explanations

1. Duration of Mechanical Ventilation; 2. ICU Length of Stay; 3. Delirium; 4. Hypotension; 5. Bradycardia; 6. Mortality; 7. Hospital Length of Stay; 8. Proportion of Time at Target Sedation; 9. Percent of Patients that Required Supplemental Sedatives; 10. Hypertension; 11. Tachycardia;12. ICU cost

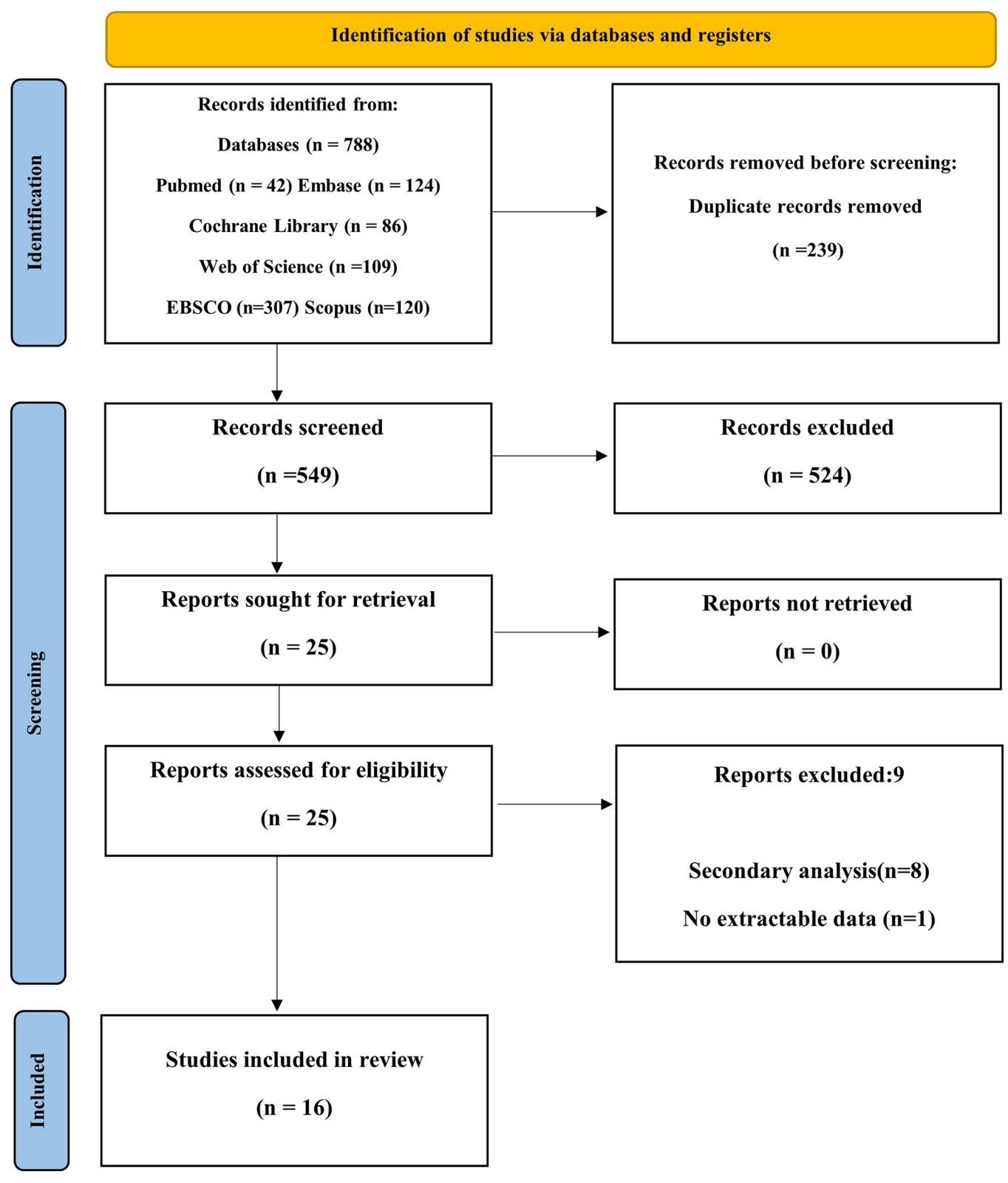

**Fig 1. Flow chart of the study selection process.**

**Table 1. Characteristics of included studies.**

| Study | Year | Race | Age (years) Dex | Age (years) Mid | APACHE II (Dex/Mid) | Number of Patients (Dex/Mid) | DEX dose | MDZ dose | Primary outcome |
|---|---|---|---|---|---|---|---|---|---|
| Zhou [32] | 2022 | China | 54.5 ±14.5 | 50.8 ±15.4 | 20±7.4/21±5.9 | 77/73 | 0.2–0.7µg/ kg/h | 0.04–0.2 mg/kg/h | 1 2 3 4 5 6 7 10 11 12 |
| Nader [33] | 2021 | Iran | 60 ±5.3 | 63±5.3 | 21.4 ± 7.4/ 20.1 ± 8.7 | 51/50 | 0.2–1µg/kg/h | 20–40µg/kg/h | 1 2 3 4 5 6 9 |
| Shu [34] | 2019 | China | 73.4 ±8.6 | 73.8 ±8.0 | 21.4±4.1/23.5 ±5.5 | 40/40 | 1µg/kg bolus then 0.2–0.7µg/ kg/h | 0.05 mg/kg bolus then 0.05–0.1 mg/ kg/h | 3 4 5 |
| Geng [35] | 2018 | China | 56.8 ±5.1 | 59.8 ±6.1 | 18.5±2.7/17 .8 ±2.2 | 42/42 | 1µg/kg bolus then 0.25–0.75µg/kg/h | 0.1 mg/kg bolus then0.1 mg/kg/h | 2 |
| Kawazoe [36, 37] | 2017 | Japan | 68 ±14.9 | 67 ±13.6 | 23±8.2/21.5±7.4 | 100/51 | 0.1–0.7µg / kg / h | 0–0.15mg/kg/h | 2 3 5 6 |
| Li [38] | 2019 | China | 43 ±15.0 | 45 ±13.0 | 20±5/21±4 | 64/62 | 0.8µg/kg/h | 0.06mg/kg/h | 1 2 3 |
| Gupta [39] | 2015 | India | 43.4 ±11.6 | 39 ±14.1 | N/A | 20/20 | 0.2–0.7µg/kg/h | 0.04–0.2 mg/kg/h | 1 |
| SriVaStaVa [40] | 2014 | India | 50.5 ±7.4 | 51.3 +8.0 | N/A | 30/30 | 1µg/kg bolus then 0.4–0.7µg/ kg/h | 0.04 mg/kg bolus then 0.08mg/ kg/ h | 1 5 |
| Shehabi [41] | 2013 | Australia | 65.0 ±15.0 | 61.6 ±17.0 | 20.2±6.2/18.6 ±8.8 | 21/16 | 0–1.5µg/kg/h | dose not specified | 2 3 6 7 |
| MacLaren [13] | 2015 | US | 58.3 ±15.3 | 57.8 ±9.3 | N/A | 11/12 | 0.15–1.5µg/kg/h | 1–10 mg/h | 1 2 3 5 6 8 11 |
| Jakob [42, 43] | 2012 | Finland | 65 ±14.1 | 65 ±14.1 | N/A | 249/251 | 0.2–1.4µg/kg/h | 0.03–0.2mg/kg/h | 1 2 3 4 5 6 7 8 9 10 11 12 |
| Huang [44] | 2012 | China | 67.4 ±8.2 | 61.5 ±7.3 | 22.6±3.9/21.4 ±4.1 | 33/29 | 0.2–0.7µg/kg/h | 0.05–0.1mg/kg/h | 2 3 4 5 6 |
| Ruokonen [45] | 2009 | Finland | 64 ±16.3 | 68 ±16.3 | N/A | 41/44 | Dex0.25–1.4µg/ kg/ h | 1–2 mg IV boluses and if insufficient0.04–0.2mg/kg/h infusion | 2 4 5 6 8 9 |
| Riker [46, 47] | 2009 | US | 61.5 ±14.8 | 62.9 ±16.8 | 19.1±7.0/18.3 ±6.2 | 244/122 | 1µg/kg bolus then 0.2–1.4µg/kg/h | 0.05 mg/kg bolus then 0.02–0.1mg/ kg/ | 2 3 4 5 6 8 9 10 11 12 |
| Maldonado [48] | 2009 | US | 58.0 ±16.0 | 60.0 ±16.0 | N/A | 40/40 | 0.4µg/kg bolus then 0.2–0.7µg/kg/h | 0.5–2 mg/h | 1 2 3 7 9 |
| Esmaoglu [49] | 2009 | Turkey | 25.1 ±4.8 | 26.8 ±7.1 | 5.1 ± 3.1/ 6.0 ± 2.7 | 20/20 | 1µg/kg bolus then 0.7µg/kg/h | 0.05 mg/kg bolus then 0.1 mg/kg/h | 2 9 |

## Risk of bias

7 trials were found to be low risk of bias according to the Cochrane risk of bias tool. There was a moderate level of quality among the included trials. We evaluated the mentioned-above items. An inadequate method of random sequence generation was applied in 1 trial and conceal allocation in 1 trial was improper. Patient or caregiver blinding was absent in 3 trials, and 4 did not follow a blinding procedure for outcome assessment. We deemed two trials to have incomplete results. Risk of bias assessment details are shown in S1 Fig in S1 File.

## Meta-analysis

Meta-analyses were performed for the intensive Care Unit Length of Stay, Duration of Mechanical Ventilation, delirium, bradycardia, hypotension, mortality and some other outcomes. In the following paragraphs, we present the results for effect sizes comparing the dexmedetomidine and midazolam.

## ICU length of stay

A total of 13 studies with 1779 patients were analyzed. There was relatively large heterogeneity (I2 = 53%, p = 0.01). The use of dexmedetomidine can reduce the ICU length of stay compared to midazolam (MD = -2.25 days, 95%CI = -2.94, -1.57, p<0.0001, moderate certainty, S1 Table in S1 File and Fig 2A). The sensitivity analyses excluding high risk of bias or small sample size studies, as well as the subgroup analysis, indicated no effect modification, which proved the robustness of our results (Fig 3A and S12-S17 Figs in S1 File). In trial sequential analysis, the AIS was larger than the RIS. The cumulative Z curves crossed both the conventional boundary and the benefit boundary, so the conclusion is conclusive and favors dexmedetomidine (Fig 4A). By employing Egger's test, publication bias was not detected (P = 0.938; S2 Fig in S1 File).

## Delirium

12 studies (n = 1738) explored the difference between dexmedetomidine and midazolam in the incidence of delirium. The studies displayed statistical heterogeneity (I2 = 51%, p = 0.03). When dexmedetomidine was used instead of midazolam, the percentage of delirious patients was lower (RR = 0.63, 95%CI = 0.50, 0.81, p = 0.0002, moderate certainty, S1 Table in S1 File and Fig 2C). The results of the subgroup analysis and sensitivity analysis suggested no effect modification (Fig 3C and S18-S23 Figs in S1 File). Additionally, we performed a meta-regression analysis to determine the risk of delirium based on age, but our results did not reach statistical significance (P = 0.213, S3 Fig in S1 File). Although the cumulative Z curves in TSA showed that the required amount of information was not enough, the Z curve crossed traditional and benefit boundaries, indicating a true positive result in favor of dexmedetomidine usage (Fig 4C). This analysis did not exhibit publication bias according to Egger's test (P = 0.246, S2C Fig in S1 File).

## Duration of mechanical ventilation

Two sedatives were compared in 11 trials (1712 patients) in terms of duration of mechanical ventilation. The I2 of 97% indicated substantial heterogeneity. Our analysis found that dexmedetomidine was associated with shorter mechanical ventilation (MD = -0.83 h, 95%CI = -1.24, -0.43, p<0.0001, very low certainty, S1 Table in S1 File and Fig 2B). It was unclear whether dexmedetomidine would reduce the duration of mechanical ventilation among long-term sedation patients or North Americans in the subgroup analysis. Significant subgroup differences were detected in subgroup analyses by age and APACHE II, but the results of the subgroup were consistent with those of the primary analysis. Sensitivity analyses also did not alter the conclusions (Fig 3C and S24-S29 Figs in S1 File). The cumulative Z curve exceeded the monitoring boundary despite the information size not reaching the required level, so the outcome of TSA demonstrated that the result was conclusive (Fig 4B). With Egger's test, there were no indications of publication bias (P = 0.200; S2B Fig in S1 File).

## Bradycardia

The pooled estimate of 10 RCTs (n = 1578 patients) showed that a high risk of bradycardia was associated with dexmedetomidine when compared to midazolam (RR 2.21, 95% CI 1.31, 3.73, p = 0.003, high certainty, S1 Table in S1 File and Fig 2D). The heterogeneity among studies was low (I2 = 42%, p = 0.08). Neither subgroup analysis nor sensitivity analysis suggested effect modification (Fig 3D and S30-S35 Figs in S1 File). While the cumulative Z curve did not reach the required information size, it crossed both the conventional and harm boundaries,

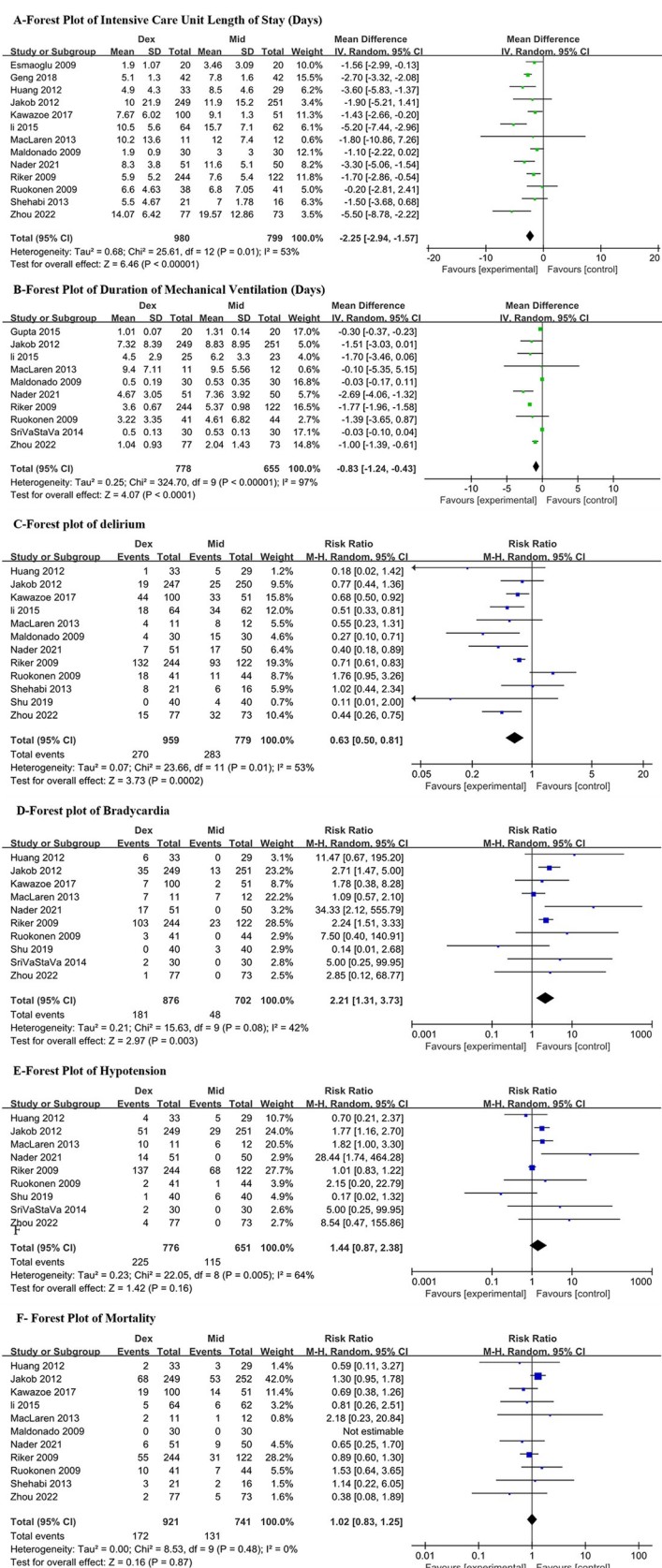

**Fig 2. Forest plot.**

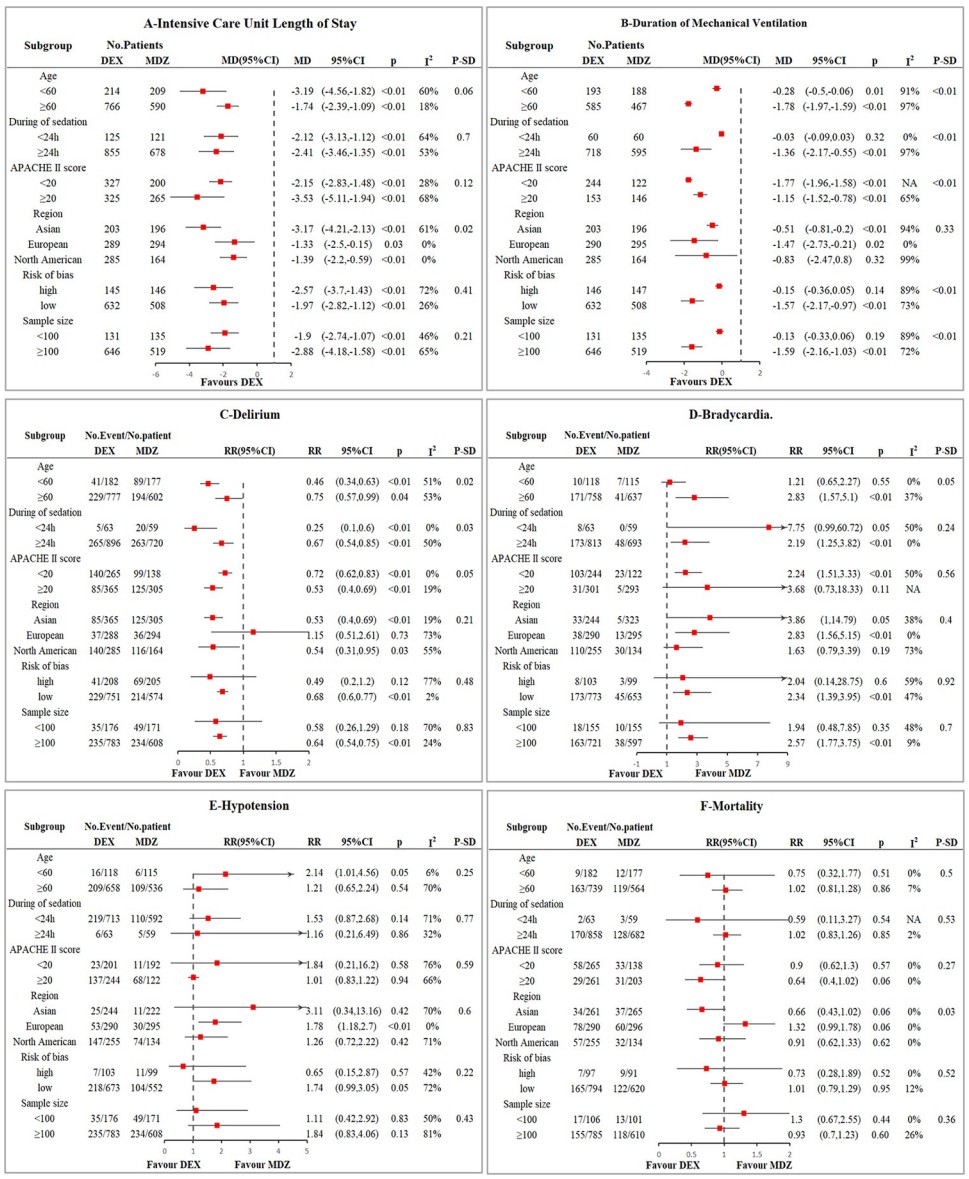

**Fig 3. Subgroup analysis.**

concluding the TSA conclusion. (Fig 4D). Egger's test revealed no publication bias (P = 0.507, S2D Fig in S1 File). However, according to the analysis of two trials (n = 428), there was no increased risk of bradycardia requiring intervention (RR 1.98, 95% CI 0.14, 28.72, S1 Table in S1 File and S10 Fig in S1 File).

## Hypotension

In 9 studies (n = 1427 patients), hypotension was included in their evaluation index. We observed heterogeneity between studies (I2 = 64%, p = 0.005). Meta analysis indicated that dexmedetomidine did not increase the risk of hypotension (RR = 1.39, 95%CI = 0.84, 2.32, P = 0.2, low certainty, S1 Table in S1 File and Fig 2E). Subgroup analysis and sensitivity analysis did not alter the primary conclusions (Fig 3E and S36-S41 Figs in S1 File). TSA conclusion

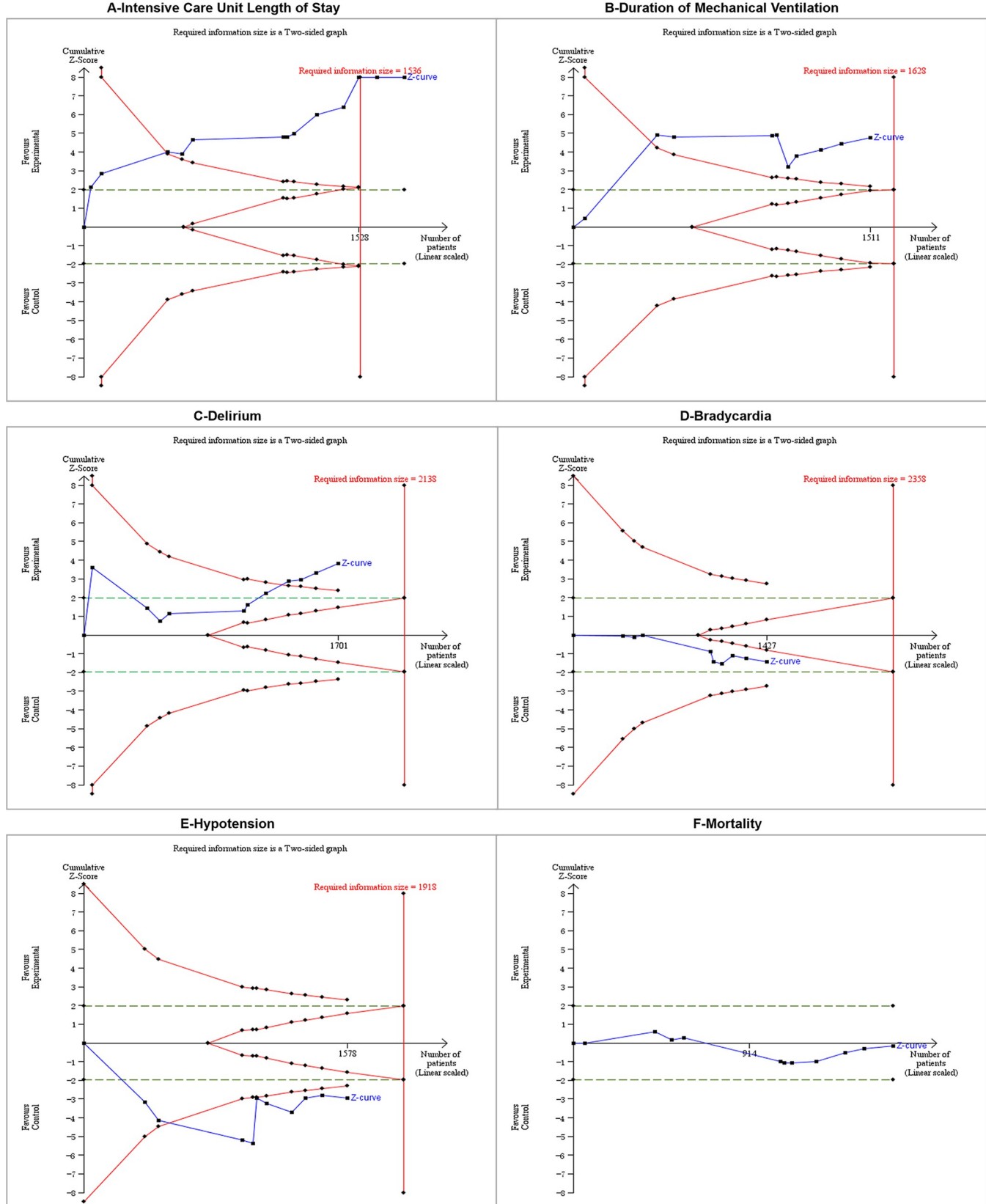

**Fig 4. Trial sequential analysis.**

is inconclusive, since the cumulative Z curves did not cross any boundaries and the AIS is smaller than the RIS, so further study is necessary to uncover firm evidence (Fig 4E). The Egger's test (P = 0.239) indicated no publication bias (S2E Fig in S1 File). The results from 5 trials (n = 609) showed no statistically significant increase in the risk of hypotension requiring intervention (RR 1.45, 95% CI 0.70, 3.00; S9 Fig in S1 File).

## Mortality

To explore the difference in mortality between dexmedetomidine and midazolam, we comprehensively analyzed 9 literatures (n = 1662 patients) Heterogeneity among these studies was low (I2 = 0%, p = 0.48). Our analysis revealed that compared with midazolam, dexmedetomidine did not show an advantage in reducing patient mortality (RR = 0.99, 95%CI = 0.81, 1.21, P = 0.92, high certainty, S1 Table in S1 File and Fig 2F). According to the results of the subgroup and sensitivity analyses, the primary analysis was reliable (Fig 3F and S42-S47 Figs in S1 File). The TSA conclusion is inconclusion. The Z curves crossed no boundaries, so further research may be needed to confirm this conclusion (Fig 4F). Based on Egger's test (P = 0.325), publication bias was not detected (S2F Fig in S1 File).

## Other outcomes

4 trials (n = 972) included the outcome of proportion of time at target sedation. Dexmedetomidine was the same as midazolam in terms of sedative effects (MD = -0.91, 95%CI = -3.90, 5.72, p = 0.71, low certainty, S1 Table in S1 File and S4 Fig in S1 File). This had also been demonstrated with regard to the outcome of using additional sedatives in 6 trials (n = 1149, RR = 1.04, 95%CI = 0.86, 1.26, p = 0.67, low certainty, S1 Table in S1 File and S5 Fig in S1 File). 4 trials included the length of the hospital stay(n = 747), there was no difference in the effects of the two sedatives on this outcome (MD = 0.11, 95%CI = -5.90, 6.11, p = 0.97, very low certainty, S1 Table in S1 File and S8 Fig in S1 File). 3 trials (n = 1016) reported hypertension. As with hypotension, the incidence of hypertension did not differ between the two sedatives (n = 1016, RR = 0.98, 95%CI = 0.74, 1.29, p = 0.87, low certainty, S1 Table in S1 File and S7 Fig in S1 File). However, midazolam was associated with an increased risk of tachycardia, according to the reports in four trials (n = 1039, RR = 0.73, 95%CI = 0.51, 1.04, p = 0.08, S1 Table in S1 File and S6 Fig in S1 File).

## Discussion

Sedation has always been a challenge in mechanical ventilation in the ICU. Our meta-analysis combined the latest evidence to analyze the merits of dexmedetomidine versus midazolam in critically ill patients, using trial sequence analysis and a GEADE system to verify the reliability of our results. Based on the results of our systematic review and meta-analysis of 16 trials including 1,998 critically ill patients, we find high certainty evidence that dexmedetomidine increases bradycardia risk compared with midazolam. It was moderately certain that dexmedetomidine reduced ICU hospitalization and delirium incidences, but did not improve survival. Low evidence indicated that it did not increase hypotension risks. Evidence that dexmedetomidine reduced the duration of mechanical ventilation was of very low certainty.

In patients with delirium, prolonged mechanical ventilation, hospital stays, increased medical costs and mortality [50–52]. There is a significant increase in the incidence of cognitive impairment during long-term follow-up of patients with delirium [53]. PADIS noted that benzodiazepine use is a modifiable factor associated with delirium [3], but several recent studies have also shown that there does not appear to be a relationship between short-acting benzodiazepines and the incidence of delirium [54–56]. Previous studies have also indicated that

midazolam seems to be unrelated to the incidence of delirium [57]. Interestingly, midazolam had also been reported to have a brain protective effect [58, 59]. In contrast to the embarrassment of midazolam, dexmedetomidine was widely believed to have delirium prevention effects [17]. In fact, this was recently questioned in a large, randomized clinical trials [60]. We compared the two sedatives in our meta-analysis. Moderate evidence was found that patients on dexmedetomidine did have a lower incidence of delirium than on midazolam. Subgroup analysis by region, APACHE II score and duration of sedation suggested no effect modification. The delirium prophylactic effect of dexmedetomidine may relate to these aspects. Dexmedetomidine leads to a lower depth of sedation [61], provides a more natural sleep-like sedation pattern [62]. However, in the subgroup analyses by age, dexmedetomidine showed a more obvious advantage over midazolam in patients aged under 60 years old. In the meta-regression analysis, we found that neither age nor baseline APACHE II score was associated with delirium efficacy, which is similar to recent studies [63, 64]. We have the following assumptions for this result. PADIS points out that increasing age is a nonmodifiable risk factor for an association with delirium [3]. Previous studies also indicated that midazolam was a risk factor for delirium. Therefore, in the face of these patients, caregivers may increase the priority of delirium monitoring and prevention, thus reducing the incidence of delirium in the midazolam group. In particular, the odds of delirium are similar between non-benzodiazepines and intermittent bolus dosing benzodiazepines [65].Many studies have also shown that delirium prevention will depend on the implementation of non-drug interventions, which have shown the greatest potential for success [66–68]. A comfortable patient without psychosis needs a suitable balance between pain medication synergy and other effective modes of non-medication management, anti-anxiety, and restorative sleep, such as family involvement. Delirium should not be a problem in the ICU in the future [69].

Sedative choice can have a potential impact on overall cost per episode [70]. In the present study, Dexmedetomidine on ICU cost showed cost-effectiveness. Although dexmedetomidine is more expensive than midazolam, we observed statistically significant reductions in ICU length of stay and mechanical ventilation time in patients sedated with dexmedetomidine. Subgroup analysis and sensitivity analysis proved that the results were reliable. Therefore, the total cost was cheaper instead. Prolonged mechanical ventilation or ICU hospitalization increased the risk of delirium and other adverse events, and increased the financial burden and emotional stress of the patient. Post-intensive care syndrome (PICS) seriously reduces the quality of life after patient discharge [71]. Negative ICU experiences and delirium were considered major risk factors [72]. Dexmedetomidine can improve sleep quality [61], decrease rates of psychological impairment, and thereby significantly lower rates of PICS [73].

Our meta-analysis found no difference between two sedatives in maintaining patient target sedation rates and no difference in the need for additional sedatives. However, dexmedetomidine has the biphasic effects of temporarily raising blood pressure through the transient constriction of peripheral blood vessels [74]. When administered in a slow manner, sedation was induced by the activation of the Alpha-2 receptor. Inhibition of sympathetic nerve activity and an increase in cardiac vagus nerve activity may lead to hypotension and bradycardia [75]. Our meta-analysis suggested that dexmedetomidine results in little to no difference from midazolam in hypotension, but there is an increasing tendency. A higher risk of bradycardia was associated with dexmedetomidine. However, there was no difference in hypotension and bradycardia requiring intervention, which was consistent with previous studies [17, 76]. Subgroup analysis and sensitivity analysis also demonstrated no modification. Furthermore, previous studies had shown that dexmedetomidine has a higher risk of hypertension than midazolam [77], nor was it observed in our meta-analysis. Although midazolam is thought to have a negligible effect on hemodynamics [78]. We found an increased tendency to

tachycardia in the midazolam group. Although dexmedetomidine may increase the risk of hypotension and bradycardia, it is closely related to the loading dose and infusion speed [79, 80]. Therefore, these adverse reactions can be prevented and reversed. In fact, many clinicians tend to avoid loading dose administration, especially in critically ill patients [81]. Therefore, the correct dose and method of dexmedetomidine can completely avoid bradycardia and hypotension.

Several limitations were also observed in this meta-analysis. First, we wanted to include the risk of self-extubation and delirium-free days as our outcome indicators, but these outcomes were not described in the included RCTS. There were too few trials describing some outcomes, which affects the level of evidence for these outcomes, and we cannot detect publication bias. Due to the high heterogeneity of some outcome indicators, the certainty of outcome is reduced. Second, most of the ICU types in the trial were mixed, so the subgroup analysis we planned to compare the medical and surgical populations was not available. Some subgroup results lacked individual patient data, and we were unable to conduct meta-regression of APACHE II score and patient weight. Due to the small number of contribution trials and participants, there was insufficient persuasion for specific subgroups.

## Conclusion

In conclusion, combined with evidence from recent studies, it suggests that dexmedetomidine is comparable to midazolam in sedation, with lower delirium, shorter mechanical ventilation duration, and shorter ICU stay. In spite of this, it may increase the risk of adverse events such as hypotension and bradycardia, and does not improve patient survival.

## Supporting information

**S1 File.**
(DOCX)

**S1 Table. PRISMA checklist.**
(DOCX)

## Author Contributions

**Conceptualization:** Jiaxuan Wen, Xueying Ding.

**Data curation:** Jiaxuan Wen, Xueying Ding, Xin Liu.

**Formal analysis:** Jiaxuan Wen, Chen Liu, Wenyu Jiang, Yingrui Xu.

**Methodology:** Chen Liu, Xiuhong Wei.

**Supervision:** Chen Liu, Xiuhong Wei.

**Writing – original draft:** Jiaxuan Wen.

**Writing – review & editing:** Jiaxuan Wen, Xin Liu.

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
