## [Decision Letter · Decision Letter 0]

11 Oct 2023

PONE-D-23-16182A comparation of dexmedetomidine and midazolam for sedation in patients with mechanical ventilation in ICU: a systematic review and meta-analysisPLOS ONE

Dear Dr. Liu,

Thank you for submitting your manuscript to PLOS ONE. After careful consideration, we feel that it has merit but does not fully meet PLOS ONE’s publication criteria as it currently stands. Therefore, we invite you to submit a revised version of the manuscript that addresses the points raised during the review process.

We look forward to receiving your revised manuscript.

Kind regards,

Ioannis Savvas, DVM, Ph.D.

Academic Editor

PLOS ONE

Journal Requirements:

Did you know that depositing data in a repository is associated with up to a 25% citation advantage (https://doi.org/10.1371/journal.pone.0230416)? If you’ve not already done so, consider depositing your raw data in a repository to ensure your work is read, appreciated and cited by the largest possible audience. You’ll also earn an Accessible Data icon on your published paper if you deposit your data in any participating repository (https://plos.org/open-science/open-data/#accessible-data).

Reviewers' comments:

Reviewer's Responses to Questions

**Comments to the Author**

1. Is the manuscript technically sound, and do the data support the conclusions?

Reviewer #1: Yes

Reviewer #2: Yes

2. Has the statistical analysis been performed appropriately and rigorously? 

Reviewer #1: Yes

Reviewer #2: Yes

3. Have the authors made all data underlying the findings in their manuscript fully available?

Reviewer #1: Yes

Reviewer #2: Yes

4. Is the manuscript presented in an intelligible fashion and written in standard English?

Reviewer #1: Yes

Reviewer #2: Yes

5. Review Comments to the Author

Reviewer #1: The authors conducted a research study titled "A Comparison of Dexmedetomidine and Midazolam for Sedation in Patients with Mechanical Ventilation in ICU: A Systematic Review and Meta-Analysis." This study addresses a critical clinical issue, and the authors have executed a well-organized and meticulously designed systematic review and meta-analysis. However, there are a few comments that need to be addressed:

Q1: The study's cutoff date was October 18th, 2022. The authors should consider performing an update of their research in accordance with PRISMA guidelines.

Q2: The exclusion criteria stating "Publications without an outcome of interest" may not be considered a valid exclusion criterion, as per the Cochrane Handbook of Intervention. The sentence should be revised to align with established criteria, such as those mentioned in the question.

Q3: In Table 1, the measurement unit for age should be included for clarity.

Q4: In Figure 1, it is advisable to provide reasons for not retrieving the two reports during the screening process. Additionally, the exclusion of 46 papers after full-text assessment raises questions. It's essential to clarify how these papers, including 22 publications that are review/meta-analyses and 15 unrelated to the research question or were animal experiments, were not excluded during title/abstract screening.

Q5: If the Forest plot is generated using RevMan, it is suggested to add labels for both groups rather than using generic "experiment/control" labels, as this can be unclear and less straightforward.

Q6: Furthermore, it is recommended to add labels for each group in the Forest plot for better clarity and understanding.

Reviewer #2: The paper is well written and complete.

The topic is interesting and methods are accurate.

I have only a suggestion:

Abstract:

lenght of stay in ICU should be added in the results because it is reported in the conclusions.

6. PLOS authors have the option to publish the peer review history of their article (what does this mean?). If published, this will include your full peer review and any attached files.

Reviewer #1: **Yes: **Cho-Hao, Lee

Reviewer #2: **Yes: **Angela Amigoni

---

## [Author Response · Author response to Decision Letter 0]

17 Oct 2023

October 17, 2023

Response for manuscript PONE-D-23-16182 “A comparation of dexmedetomidine and midazolam for sedation in patients with mechanical ventilation in ICU: a systematic review and meta-analysis”

Dear Editors:

Thank you for providing us with such a great opportunity to submit a revised version of our manuscript. We appreciate your warm help with our manuscript. Meanwhile, we would like to express our sincere gratitude to all reviewers for their detailed and constructive comments on our manuscript. According to those helpful suggestions, we have revised the manuscript to make our results convincing. Revised portions are marked in red on the paper. The main corrections in the paper and the responses to the reviewers' comments are as follows. 

At the same time, we ensure that the manuscript meets PLOS ONE's style requirements. The first author's ORCID ID is 0000-0002-5625-7134, and the corresponding author's ORCID ID is 0009-0009-9798-407X. we have checked all the references to ensure that they are complete and correct.

Thank you again for your time and help to our manuscript. We hope you will be satisfied with the revised version and look forward to hearing from you.

Your sincerely,

Corresponding author

Xin Liu

Encl. Responses to the comments from Reviewer 1 and 2.

Reply to Reviewer #1

Dear Reviewer,

Thank you very much for your time spent in reviewing our manuscript and for your encouraging comments on its merits. After careful consideration, we have further revised the article. We hope that you will be more satisfied with the revised version.

Comments 1:

The study's cutoff date was October 18th, 2022. The authors should consider performing an update of their research in accordance with PRISMA guidelines.”

Response 1:

Thank you very much for your affirmation of this article. Following the guidance of the PRISMA guidelines, we conducted an electronic search for recent trials to perform an update of our research. 

[Relative revision can be found in the Materials and Methods part, line 73-78]

Comments 2:

The exclusion criteria stating "Publications without an outcome of interest" may not be considered a valid exclusion criterion, as per the Cochrane Handbook of Intervention. The sentence should be revised to align with established criteria, such as those mentioned in the question.

Response 2:

Your suggestions really mean a lot to us. we have revised our eligibility criteria under the guidance of the Cochrane Handbook of Intervention. Thank you very much.

[Relative revision can be found in the Materials and Methods part, line 84-86]

Comments3:

In Table 1, the measurement unit for age should be included for clarity. 

Response 3:

Thanks for your valuable comments. We have added the measurement unit for age in Table 1.

[Relative revision can be found in Table 1].

Comments 4:

In Figure 1, it is advisable to provide reasons for not retrieving the two reports during the screening process. Additionally, the exclusion of 46 papers after full-text assessment raises questions. It's essential to clarify how these papers, including 22 publications that are review/meta-analyses and 15 unrelated to the research question or were animal experiments, were not excluded during title/abstract screening.

Response 4:

Thanks for your valuable comments. We used Endnote X9 software to remove duplicate reports. Two of the duplicate reports were not automatically deleted. When we retrieved full papers, we found these two records and removed them. After discussion, we believed that we had caused a misunderstanding, these two reports should be excluded after the records screening stage. I am ashamed that I did not catch the mistake in time. In order to improve recall ratio, we screened the references mentioned in meta-analyses and any other papers that may be relevant to our study. These articles were gotten the full text and read carefully. This led the author of the flow chart to mistakenly include these articles in the full-text assessment. According to the latest electronic search results, we have carefully revised the flow chart.

[Relative revision can be found in Results part, line 131-136 and Figure 1]

Comments 5:

If the Forest plot is generated using RevMan, it is suggested to add labels for both groups rather than using generic "experiment/control" labels, as this can be unclear and less straightforward.

Response 5:

Thanks for your good advice. We added the label for both groups using RevMan.

[Relative revision can be found in Results part, Figure 2]

Comments 6:

Furthermore, it is recommended to add labels for each group in the Forest plot for better clarity and understanding.

Response 6:

Thanks for your valuable comments. We have added labels for each group and optimized each group of figures for better clarity and understanding.

[Relative revision can be found in Results part, Figure 2-4]

Thank you very much for your affirmation and suggestions on this article. We hope the revised version will make you even more satisfied. If there is anything that needs to be modified, please do not hesitate to point it out. We are more than happy to make any further changes that improve the article.

Sincerely,

Corresponding author

Xin Liu

Reply to Reviewer #2

Dear Reviewer,

Thank you very much for your time spent in reviewing our manuscript and for your encouraging comments on its merits. After careful consideration, we have further revised the article. We hope that you will be more satisfied with the revised version.

Comments:

Abstract:

length of stay in ICU should be added in the results because it is reported in the conclusions.

Response:

We have added the information in the manuscript. We hope you will find this revised version more satisfactory. We are more than happy to make any further changes that will improve the article and facilitate successful publication.

[Relative revision can be found in the abstract part, line 28-34]

Sincerely,

Corresponding author

Xin Liu

---

## [Decision Letter · Decision Letter 1]

30 Oct 2023

A comparation of dexmedetomidine and midazolam for sedation in patients with mechanical ventilation in ICU: a systematic review and meta-analysis

PONE-D-23-16182R1

Dear Dr. Liu,

We’re pleased to inform you that your manuscript has been judged scientifically suitable for publication and will be formally accepted for publication once it meets all outstanding technical requirements.

Kind regards,

Ioannis Savvas, DVM, Ph.D.

Academic Editor

PLOS ONE

Additional Editor Comments (optional):

Reviewers' comments:

Reviewer's Responses to Questions

**Comments to the Author**

1. If the authors have adequately addressed your comments raised in a previous round of review and you feel that this manuscript is now acceptable for publication, you may indicate that here to bypass the “Comments to the Author” section, enter your conflict of interest statement in the “Confidential to Editor” section, and submit your "Accept" recommendation.

Reviewer #1: All comments have been addressed

Reviewer #2: All comments have been addressed

2. Is the manuscript technically sound, and do the data support the conclusions?

Reviewer #1: Yes

Reviewer #2: Yes

3. Has the statistical analysis been performed appropriately and rigorously? 

Reviewer #1: Yes

Reviewer #2: Yes

4. Have the authors made all data underlying the findings in their manuscript fully available?

Reviewer #1: Yes

Reviewer #2: Yes

5. Is the manuscript presented in an intelligible fashion and written in standard English?

Reviewer #1: Yes

Reviewer #2: Yes

6. Review Comments to the Author

Reviewer #1: The authors has responded to all my questions, this is a well-organized research. The quality is enough to be published

Reviewer #2: I don't have further comments to add. All the input have been followed. This is a well written paper.

7. PLOS authors have the option to publish the peer review history of their article (what does this mean?). If published, this will include your full peer review and any attached files.

Reviewer #1: **Yes: **Cho-Hao, Lee

Reviewer #2: No

---

## [Editor Report · Acceptance letter]

3 Nov 2023

PONE-D-23-16182R1 

A comparation of dexmedetomidine and midazolam for sedation in patients with mechanical ventilation in ICU: a systematic review and meta-analysis 

Dear Dr. Liu:

I'm pleased to inform you that your manuscript has been deemed suitable for publication in PLOS ONE. Congratulations! Your manuscript is now with our production department. 

Kind regards, 

on behalf of

Prof. Ioannis Savvas 

Academic Editor

PLOS ONE